# Study on Green Controllable Preparation of Coal Gangue-Based 13-X Molecular Sieves and Its $CO_2$ Capture Application

Dawei Yi [1], Huiling Du [1], Yefei Li [2,*], Yimin Gao [2,*], Sifan Liu [1,3], Boyang Xu [1], Haoqi Huang [1] and Le Kang [1]

1   College of Materials Science and Engineering, Xi'an University of Science and Technology, Xi'an 710054, China; yidawei19820608@163.com (D.Y.); hldu@xust.edu.cn (H.D.); sifan433320@163.com (S.L.); xuboyanglwz@163.com (B.X.); g2huanghaoqi@163.com (H.H.); kangle20140805@126.com (L.K.)
2   State Key Laboratory for Mechanical Behavior of Materials, College of Materials Science and Engineering, Xi'an Jiaotong University, Xi'an 710049, China
3   Shaanxi Longmen Iron and Steel Co., Ltd., Hancheng 715400, China
*   Correspondence: liyefei@xjtu.edu.cn (Y.L.); ymgao@xjtu.edu.cn (Y.G.)

**Abstract:** Carbon dioxide emissions are the primary and most direct contributor to global warming, posing a significant hazard to both the environment and human health. In response to this challenge, there has been a growing interest in the development of effective carbon capture technologies. This study involved the synthesis of 13-X molecular sieve porous materials using solid waste coal gangue as a source of silicon and aluminum. The synthesis process involved the controlled utilization of an "alkali fusion-hydrothermal" reaction system. The resulting materials were characterized for their structure, morphology, and crystal composition using X-ray diffraction and field emission scanning electron microscopy. These 13-X molecular sieve materials were employed as adsorbents to capture carbon dioxide gas, and their adsorption performance was investigated. The findings indicated that the 13-X molecular sieve materials possess uniform pores and complete crystalline morphologies, and they exhibited an adsorption capacity of 1.82 mmol/g for carbon dioxide at 0 °C. Consequently, this study not only converted solid waste gangue into high-value products but also demonstrated effective atmospheric carbon dioxide capture, suggesting that gangue-based 13-X molecular sieves may serve as a potential candidate for carbon capture.

**Keywords:** coal gangue; hydrothermal synthesis; 13-X molecular sieve; $CO_2$ capture

## 1. Introduction

One of the primary contributors to global warming is the substantial volume of carbon dioxide emissions resulting from human activities, such as the combustion of coal and natural gas for electricity generation [1]. Over the past few centuries, atmospheric $CO_2$ levels have risen dramatically to 418 ppm, doubling global $CO_2$ emissions compared to the 1970s. This has led to the production of approximately 3.2 billion tons of $CO_2$ annually and a 1.5 °C increase in global average temperatures, exerting significant pressure on environmental management [2–5]. Records show that even within indoor spaces, $CO_2$ concentrations continue to rise, posing long-term threats to human health, travel, food, security, water resources, the rising sea level, and biodiversity [6–8]. Consequently, there is a need to control $CO_2$ levels and research effective methods for capturing $CO_2$ from the atmosphere, with adsorption technology emerging as a promising avenue due to its versatility and applicability [9,10]. Currently, available adsorbents include activated carbon [11,12], graphene [13,14], metal oxides [15], metal–organic frameworks [16,17], and zeolites [18–20], all possessing $CO_2$ adsorption sites within their crystal structures. In light of these pressing challenges, it is imperative to urgently address the rising atmospheric $CO_2$ concentrations.

Molecular sieves, also known as zeolites, are crystalline silica–aluminates composed of $[SiO_4]^{4-}$ and $[AlO_4]^{5-}$ tetrahedra, forming three-dimensional spatial network structures

with abundant fixed-pore size channels or cage structures [21–23]. These unique pore structures and large surface areas make them valuable in the realm of adsorbents [24–26]. As coal remains a major energy source, a substantial amount of coal gangue is produced during coal mining and washing processes. It is estimated that global coal production and consumption reached 8.1 billion tons and 15.7 billion tons, respectively, in 2019 since the 18th century [27,28]. Coal gangue (CG), a byproduct of coal mining, is associated with coal and yields 100–150 kg of gangue for every ton of coal produced [29]. In China, the cumulative accumulation of coal gangue has reached 4.5 billion tons due to years of coal mining, with an annual growth rate of 300–350 million tons [30]. This not only occupies significant land but also leads to air and water pollution, resulting in ecological degradation and health risks [31–33]. Given that coal gangue primarily contains silica and aluminum clay minerals [34,35], it is an ideal raw material for high-purity molecular sieve synthesis, addressing the challenge of coal gangue treatment and disposal.

Addressing these challenges, this study focused on the preparation of gangue-based 13-X molecular sieve porous materials using coal gangue as a source of silicon and aluminum. Optimal synthesis conditions for 13-X molecular sieves were explored, including Si/Al ratio (PCG to $NaAlO_2$ mass ratio), alkalinity (NaOH(aq) concentration), crystallization temperature, and hydrothermal time. The resulting molecular sieve porous materials exhibited excellent $CO_2$ adsorption performance, offering a sustainable solution to both coal gangue accumulation and the increase in atmospheric $CO_2$ concentrations, promoting circular economy and waste management objectives.

## 2. Experiment

### 2.1. Materials

In the experiment, coal gangue was used as raw material to provide silicon and aluminum sources, and it was provided by China Shaanxi Shendong Coal Mining Group of Shaanxi Province, China. Sodium hydroxide (NaOH, AR, $\geq$99.70%) and sodium meta-aluminate ($NaAlO_2$, AR, $\geq$99.70%) were purchased from Tianjin Damao Chemical Reagent Factory. The experimental water was homemade deionized water prepared in the laboratory. All chemical reagents were used as they were without further purification.

### 2.2. Methods

#### 2.2.1. Synthesis of Gangue-Based 13-X Molecular Sieve

The crystal structure of inert amorphous quartz and kaolinite in raw coal gangue is decomposed in the strong alkali environment and participates in the chemical reaction to produce soluble aluminosilicate $Na_2SiO_3$ and $NaAlO_2$, eliminating the influence of quartz and kaolinite on the purity of the product. Through the preliminary experiment, the best ash-to-alkali ratio of pre-treated coal gangue has been explored, and the treatment methods are as follows: The raw coal gangue powder was mixed with solid NaOH particles by mass ratio of 1:0.6 in a mortar and ground thoroughly, placed in a muffle furnace, and calcined at 750 °C under air conditions for 4 h to obtain pretreatment gangue (PCG). PCG (7.0 g) and $NaAlO_2$ (1.17 g) were added to NaOH solution (0.3 mol·$L^{-1}$, 75 mL). Then, the stirred mixture was subjected to hydrothermal synthesis reaction at 80 °C for 10 h in stainless steel PTFE reactor liner; after the reactor cooled naturally, the hydrothermal products were filtered, washed, and dried to synthesize 13-X molecular sieve, and Figure 1 shows the flow chart of molecular sieve preparation.

In the context of adsorbing airborne substances, most adsorbents are in granular form, necessitating the transformation of powdered molecular sieve materials into granules [36,37]. The synthesized 13-X molecular sieve powder is initially washed to achieve a pH of approximately 7 and dried at 60 °C for 3–4 h. Subsequently, the powder is compressed into cylinders under a pressure of 20–30 MPa, and particles with diameters ranging from 1 to 3 mm are obtained through crushing and sieving. In the adsorption experiments, ultra-high purity $CO_2$ (U-Sung, 99.999%) is utilized, and the molecular sieve particles undergo evacuation at temperatures between 200 and 250 °C prior to measurement.

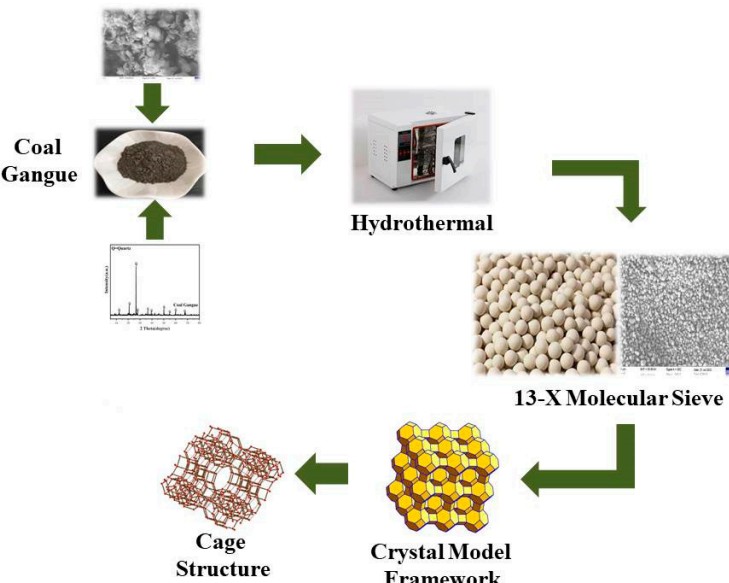

**Figure 1.** Flow chart of preparation of gangue-based 13-X molecular sieve.

2.2.2. Characterization

X-ray fluorescence spectrometer (XRF, Bruker S4 pioneer, Karlsruhe, Germany) was used to determine the chemical composition of the gangue for quantitative analysis. The crystallinity of the samples was evaluated by physical phase retrieval on a Bruker D8-FOCUS type X-ray diffractometer with a scanning speed of 5°/min and a scanning angle of 5°~80°. Field emission scanning electron microscopy (FESEM, Gemini SEM 500, Hitachi, Japan) was used to observe the microstructure and surface morphology of the samples, and the microanalysis was determined by energy spectrum analysis (EDS). The specific surface area and pore size distribution of the samples were obtained using the $N_2$ adsorption–desorption isotherm at 77 K of liquid nitrogen temperature using Micromeritics ASAP 2020 analyzer (Micromeritics, Norcross, GA, USA). The $CO_2$ gas adsorption–desorption isotherms of 13-X molecular sieve were measured on a Quantachrome version 5.21 gas adsorption analyzer.

## 3. Results and Discussion

### 3.1. Processing and Analysis of Raw Materials

3.1.1. Treatment of Raw Material Gangue

To enhance the crystallinity of the molecular sieve, coal gangue is initially crushed using a crusher and ball mill, followed by sieving through a 200 mesh sieve to obtain gangue powder. The primary components of the gangue are silicon and aluminum compounds, with negligible quantities of other compounds. Chemical composition analysis was conducted using X-ray fluorescence spectrometry (XRF, Bruker S4 pioneer, Germany), revealing that the gangue is primarily composed of $SiO_2$ and $Al_2O_3$, along with trace oxide impurities containing Ca, Mg, K, Fe, Na, and other elements, as well as some organic carbon (Table 1). The XRD pattern of the gangue indicates that its main mineral composition is quartz (Figure 2). Field emission scanning electron microscopy (FESEM) images demonstrate that the raw material consists of irregular lumps or agglomerates with non-uniform particle size distribution and a significant presence of non-crystalline particulate matter on its surface.

**Table 1.** Chemical composition of coal gangue.

| Oxide | CaO | MgO | $SiO_2$ | $Al_2O_3$ | $Fe_2O_3$ | $TiO_2$ | $K_2O$ | $Na_2O$ | Loss |
|-------|-----|-----|---------|-----------|-----------|---------|--------|---------|------|
| wt% | 0.82 | 1.57 | 61.39 | 23.76 | 4.14 | 0.79 | 2.82 | 1.62 | 2.11 |

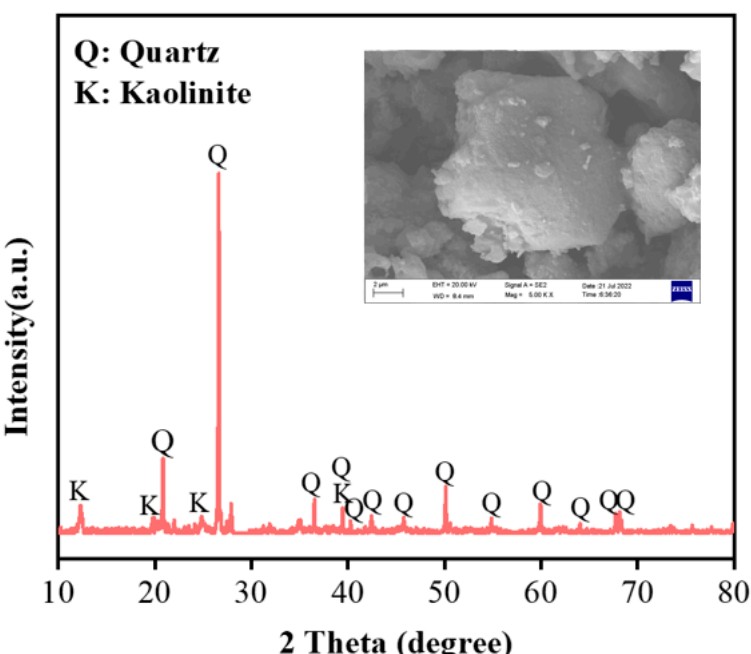

**Figure 2.** XRD pattern and FESEM pattern of coal gangue.

3.1.2. Orthogonal Experimental Design for the Synthesis of 13-X Molecular Sieve

In this study, there are many factors affecting the preparation of a 13-X molecular sieve, and an orthogonal test can conduct a comprehensive experiment on the collocation between various levels of each factor, so an orthogonal test is chosen as the research method. After the preliminary exploration experiment, the influence factors and levels of the experiment were controlled at four factors and three levels, and the orthogonal table $L_9(3^4)$ was selected, as shown in Table 2 below.

**Table 2.** Orthogonal experimental scheme.

| Serial Number | Influence Factor | | | |
|---|---|---|---|---|
| | Si/Al (PCG/NaAlO$_2$ Mass Ratio) | Alkalinity ($C_{NaOH(aq)}$)/mol·L$^{-1}$ | Crystallization Temperature/°C | Hydrothermal Time/h |
| Y1 | 4 | 0.3 | 100 | 8 |
| Y2 | 4 | 0.4 | 120 | 10 |
| Y3 | 4 | 0.5 | 80 | 12 |
| Y4 | 5 | 0.5 | 100 | 10 |
| Y5 | 5 | 0.3 | 120 | 12 |
| Y6 | 5 | 0.4 | 80 | 8 |
| Y7 | 6 | 0.4 | 100 | 12 |
| Y8 | 6 | 0.5 | 120 | 8 |
| Y9 | 6 | 0.3 | 80 | 10 |

This study explored the impact of four key factors, Si/Al ratio (PCG to NaAlO$_2$ mass ratio), alkalinity (NaOH(aq) concentration), crystallization temperature, and hydrothermal time, on the crystallinity of gangue-based 13-X molecular sieve products. The synthesized samples were denoted as Y1, Y2–Y9, etc.

*3.2. Evaluation of the Synthesized 13-X Zeolite*

3.2.1. X-ray Diffraction Analysis

The influence of the Si/Al molar ratio on the crystallization type of molecular sieve was examined. The coal gangue exhibits a high content of SiO$_2$ and Al$_2$O$_3$, with an n(Si)/n(Al) ratio of 2.20. Given that the X-type molecular sieve is considered a low-silica molecular sieve, the raw coal gangue is well-suited for the synthesis of low-silica molecular sieve

porous materials with high crystallinity. The XRD spectra of synthetic samples Y1-Y9 were compared with standard 13-X molecular sieve values (PDF#12–0246) (Figure 3). Samples Y7 and Y9 exhibited distinct peaks at 2θ of 6.090°, 9.998°, 11.696°, 15.424°, 20.073°, 23.310°, and 31.015°, indicating the presence of 13-X molecular sieve diffraction peaks. Other synthetic samples did not exhibit a 13-X molecular sieve, but the amorphous silicate content within the raw material system increased, and the quartz phase diminished as the reaction progressed. This transformation was due to the interaction of alkaline substances with the quartz crystalline phase on the gangue's surface, leading to structural alterations. These findings underscore the significance of the $SiO_2$ and $Al_2O_3$ ratio in the parent composition in determining the molecular sieve structure type.

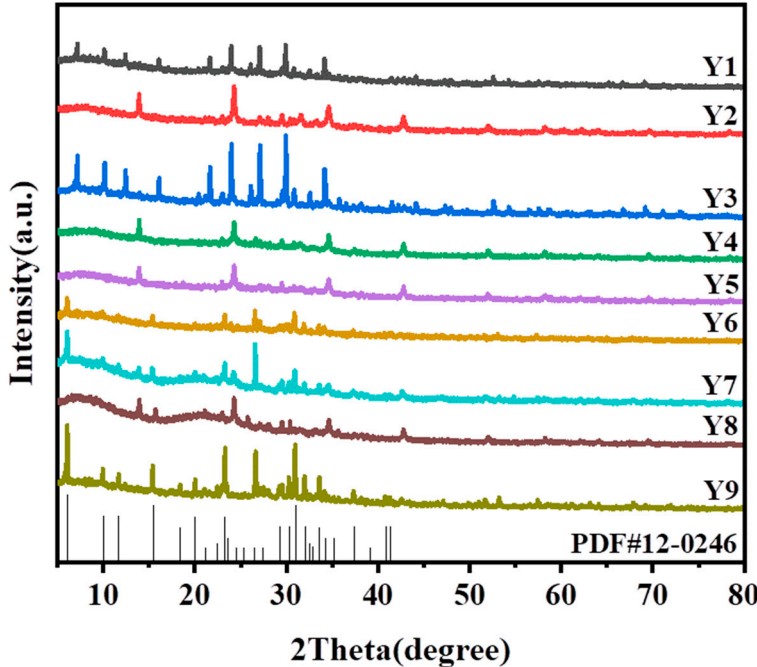

**Figure 3.** XRD patterns of the synthesized samples Y1–Y9 and 13-X standard card (bottom of the image).

The impact of alkali addition on the synthesized products was also analyzed, showing that the addition of an alkali source effectively disrupted the inert crystal structure of quartz in the gangue, enabling the dissolution of active silicoaluminate components. Alkalinity, determined by different alkali concentrations, significantly influenced the crystallinity of molecular sieves. In samples with a Si/Al molar ratio (PCG to $NaAlO_2$ mass ratio = 6), varying NaOH solution concentrations (0.4, 0.5, and 0.3 mol·L$^{-1}$) were tested. Samples Y7 and Y9, which exhibited the diffraction peaks of a standard 13-X molecular sieve, indicated that higher NaOH addition was not conducive to 13-X molecular sieve nucleation. An increase in alkalinity led to a gradual reduction in the diffraction intensity of the 13-X molecular sieve. Sample Y8 had excessively high alkalinity, resulting in a highly basic crystallization reaction process. Consequently, it hindered silicate ion polymerization and produced different structures. The synthesis of molecular sieves through alkali fusion and appropriate alkalinity adjustment treatments promoted the depolymerization of Si-Al-radical ions and facilitated zeolite grain formation and crystal growth. Different NaOH ratios yielded distinct crystal structures, with a material concentration of 0.3 mol·L$^{-1}$ providing well-developed 13-X molecular sieves. This sufficient alkalinity promoted the uniform nucleation of pure 13-X zeolites and supplemented the sodium content in the molecular sieves.

The synthesis of molecular sieves involves a process of silica–aluminate precipitation in an alkali solution, and changes in crystallization temperature affect the interaction between silica–aluminate and $[SiO_2(OH)_2]^{2-}$ and $[Al(OH)_4]^-$. To investigate the impact

of different crystallization temperatures on the synthesis of 13-X molecular sieves, three series of syntheses were conducted at 80 °C, 100 °C, and 120 °C. As shown in Figure 3, when the crystallization temperature was set at 120 °C, the formation of crystals of other structures or impurities occurred, as evidenced by the absence of standard 13-X zeolite diffraction peaks and the appearance of different crystal structures. Consequently, 13-X zeolites could not be synthesized at this temperature. When the crystallization temperature was reduced to 80 °C and 100 °C, high-purity x-type zeolites were formed. Notably, 13-X zeolites synthesized at 80 °C exhibited higher efficiency, with sample Y9 approaching the relative crystallinity of standard 13-X. Therefore, considering energy conservation and maintaining the integrity of the 13-X molecular sieve's crystal structure, 80 °C is more suitable for producing high-purity 13-X zeolites.

Hydrothermal time plays a role in the growth and formation of molecular sieve crystals, and different hydrothermal crystallization times can yield various crystal structures. The effect of hydrothermal time was studied in three series, 8 h, 10 h, and 12 h, to regulate the growth and formation of synthetic 13-X molecular sieves.

The XRD spectrum in Figure 3 reveals that at a crystallization time of 10 h, sample Y9 displayed the highest crystallinity of 13-X zeolite. Subsequently, the diffraction intensity of 13-X zeolite decreased beyond 10 h, suggesting the conversion of some X zeolites into more stable zeolites as a result of increased silicon dissolution during longer hydrothermal synthesis. Therefore, precise control of hydrothermal time is crucial in the synthesis of 13-X molecular sieves, with 10 h being considered the optimal duration for the formation of crystalline 13-X zeolites.

Taking into account the Si/Al molar ratio (PCG to $NaAlO_2$ mass ratio = 6), a NaOH solution concentration of 0.3 mol·L$^{-1}$, a crystallization temperature of 80 °C, and a hydrothermal time of 10 h, the best conditions for synthesized 13-X molecular sieves were achieved. The resulting product, Y9, exhibited sharp and complete characteristic peaks, minimal amorphous peaks, and a high degree of purity in its crystalline phase.

### 3.2.2. SEM-EDS Analysis

The electron microscope allows observation of the surface morphology and crystal size of molecular sieves, offering insights into crystal uniformity, the presence of heterocrystals, and the overall image quality. To study the crystal pattern variations, the synthesized product is initially gently ground, pretreated, gold-sprayed, and subjected to vacuum treatment. It can be seen from Figure 3 that the XRD spectrum of the synthesized sample Y1-Y9 is compared with the standard 13-X molecular sieve value (PDF#12-0246), as shown in Figure 3. The peaks of Y7 and Y9 at the 2° curves of 6.090°, 9.998°, 11.696°, 15.424°, 20.073°, 23.310°, and 31.015° indicate that the synthesis products have 13-X molecular sieve diffraction peaks at this point. Secondly, the surface morphology and crystal size of the synthesized molecular sieve samples were observed by electron microscopy. The regular octahedral structure of the 13-X molecular sieve appeared in both the Y7 and Y9 samples, but due to the difference in the level of synthetic factors in the Y7 samples, other types of molecular sieve structures were generated. Therefore, Y7 was selected for comparison and discussion with the optimal group Y9. SEM and EDS patterns of the raw material gangue, synthesized sample Y7, and the best sample Y9 are presented in Figure 4. Figure 4a,b illustrate the microscopic morphology of raw coal gangue, displaying dark gray powder after calcination. The material exhibits various shapes and sizes of lamellar structures, with lamellae stacked together in a cascade arrangement, indicating inadequate crystallization. Figure 4c presents the EDS energy spectrum analysis of raw coal gangue, revealing the main characteristic peaks corresponding to elements that constitute the octahedral crystals of the 13-X molecular sieve. Figure 4d,e show the microscopic morphology of synthesized sample Y7, characterized by an absence of octahedral structure and irregular spherical crystal morphology with uniform particle size. This indicates that the synergistic effect of high alkalinity, a long hydrothermal time, and a high temperature is not conducive to the nucleation and crystal growth of the 13-X zeolite molecular sieve. Figure 4f displays the

EDS spectrum analysis of the Y7 sample molecular sieve. In Figure 4g,h, the microscopic morphology of the best sample, Y9, reveals ortho-octahedral molecular sieve crystals with clear angles, uniform particle size and a regular overall shape [38]. The particle size is approximately 2 μm, and there is no fuzzy colloid in the electron microscope image. Figure 4i provides the EDS energy spectrum analysis of the Y9 sample, indicating that the synthesized product Y9 molecular sieve contains elements O, Na, Al, and Si with weight percentages of 35.60, 8.24, 12.05, and 17.38, respectively. The molar ratio of Si-Al is approximately 1.4, which is close to the theoretical formula of $Na_2O \cdot Al_2O_3 \cdot 2.5SiO_2 \cdot 6H_2O$. These findings affirm the high quality and crystalline purity of the synthesized products.

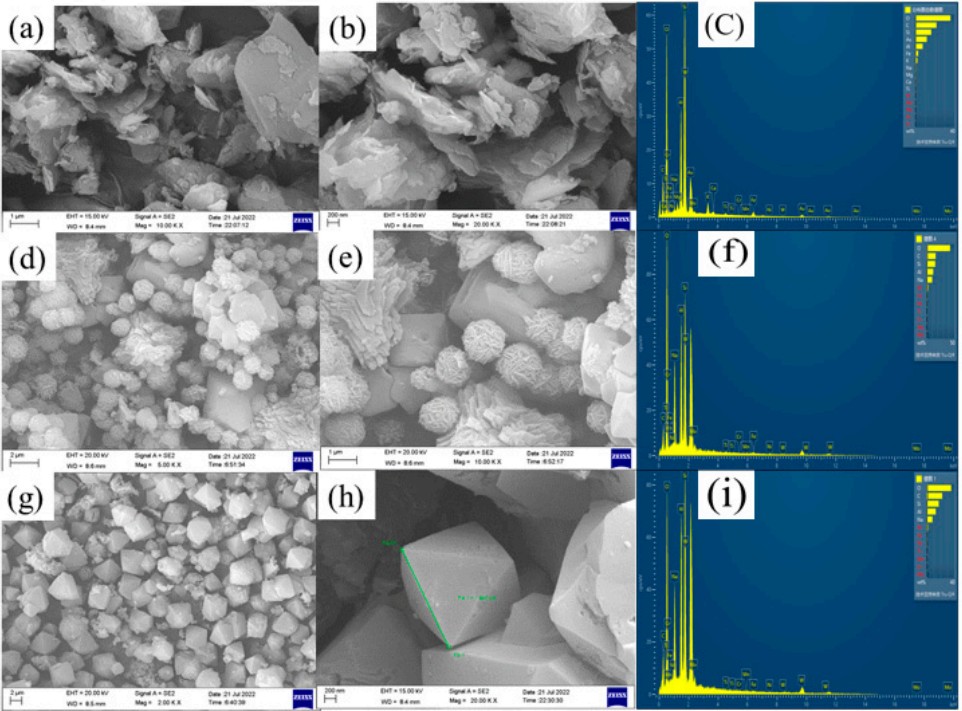

**Figure 4.** SEM patterns of raw coal gangue, (**a**,**b**); Y7 sample, (**d**,**e**); Y9 sample, (**g**,**h**). EDS patterns of raw coal gangue, (**c**); Y7 sample, (**f**); and Y9 sample, (**i**).

### 3.2.3. Specific Surface Area Analysis

The specific surface area is a crucial parameter for evaluating the adsorption capacity of molecular sieve porous materials. In general, a larger specific surface area indicates a stronger adsorption capacity. Table 3 presents the surface structure parameters of the coal gangue Y9 molecular sieve, with the BET method used to measure the surface area. The data in the table show that the BET-specific surface area of coal gangue is 2.04 $m^2/g$, with almost no adsorption capacity. After the synthesis in the alkali fusion–hydrothermal system, sample Y9's BET-specific surface area significantly increased to 377.02 $m^2/g$, accompanied by a qualitative improvement. The average pore size reached 2.09 nm, approximately 185 times that of the original gangue. These results demonstrate that the experimentally synthesized 13-X molecular sieve exhibits a high BET-specific surface area. Figure 5 presents the $N_2$ adsorption–desorption curve and pore size distribution of sample Y9, revealing increased adsorption with rising pressure. The adsorption–desorption isotherm curve resembles a typical Langmuir IV curve, indicating the presence of slit mesopores and a typical physical adsorption process. The pore size distribution of sample Y9 primarily centers on 38.8 nm mesopores, signifying a favorable pore structure. These findings confirm that the experimentally synthesized sample Y9 offers a high specific surface area and predominantly features physisorption, which is easily regenerated and recyclable, enhancing its ability to interact with adsorbents.

**Table 3.** Surface structural parameters of gangue, Y9 molecular sieves, and commercial 13-X molecular sieves.

| Sample | BET-Specific Surface Area (m²/g) | Average Pore Size (Å) | Pore Volume (m³/g) |
|---|---|---|---|
| Coal Gangue | 2.04 | — | — |
| Y9 | 377.02 | 20.87 | 0.20 |
| Commercial 13-X molecular sieve | 523.88 | 18.62 | — |

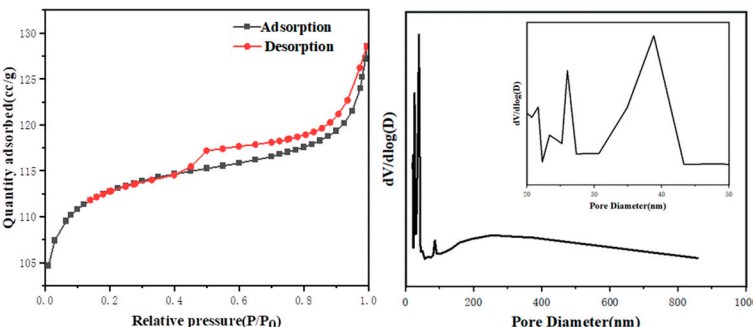

**Figure 5.** $N_2$ adsorption–desorption isotherm curve and pore size distribution of Y9 molecular sieve.

### 3.3. Measured Characteristics of $CO_2$ Adsorption by Gangue-Based 13-X Molecular Sieve

Figure 6 displays a bar graph representing the adsorption capacity of the synthesized molecular sieve for $CO_2$ at different temperatures. The gas adsorption process generally releases heat, making temperature a critical factor affecting molecular sieve gas adsorption. As depicted in Figure 6, the $CO_2$ adsorption capacity of all the synthesized molecular sieve samples, particularly Y9, diminishes as the temperature rises. The maximum $CO_2$ adsorption capacity of the 13-X molecular sieve is observed at 0 °C (273 K), reaching 1.82 mmol/g.

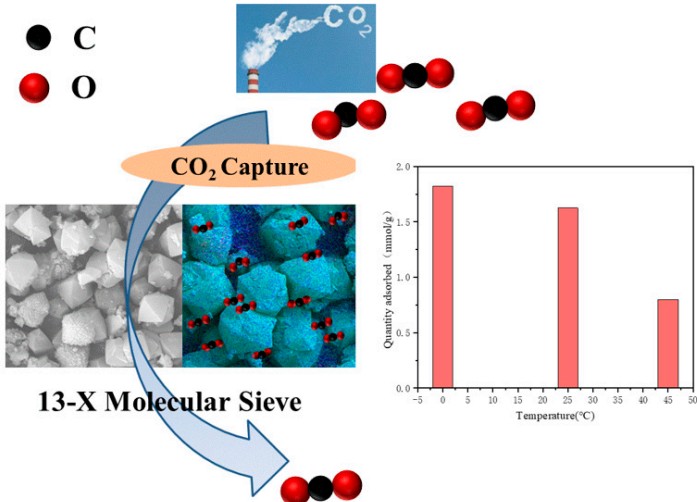

**Figure 6.** Column chart of $CO_2$ adsorption capacity of synthesized molecular sieve at different temperatures.

A number of adsorption models are commonly used to predict the adsorption of adsorbents at different pressures as a reference for further studies of the adsorption mechanism. In this study, the Langmuir Model and Dual-site Langmuir Model were used to fit the $CO_2$ adsorption isotherm.

Langmuir Model:

$$q = q_A \frac{b_A p}{1 + b_A p} \tag{1}$$

Dual-site Langmuir Model:

$$q = q_A \frac{b_A p}{1 + b_A p} + q_B \frac{b_B p}{1 + b_B p} \qquad (2)$$

In Equations (1) and (2), q is the adsorption amount, mmol/g; p is the absolute pressure, kPa; $q_A$ and $q_B$ are the saturation adsorption amounts, mmol/g; $b_A$ and $b_B$ are the affinity constants, kPa.

Figure 7 illustrates the isothermal fitting curves for $CO_2$ adsorption at various temperatures for the synthesized 13-X molecular sieve Y9, employing the Dual-site Langmuir Model for fitting. Table 4 presents the correlation coefficient values of both the Langmuir Model and the Dual-site Langmuir Model. It is evident that the Dual-site Langmuir Model offers a more accurate description of the variable pressure adsorption process for $CO_2$ gas within the synthesized 13-X molecular sieve. As the pressure gradually increases, the two adsorption sites work synergistically, causing changes in their affinity effects. Additionally, due to the mesoporous structure of the synthesized 13-X molecular sieve, adsorption predominantly takes place within the mesopores at higher relative pressures. Inside the molecular sieve skeleton, cavities contain charge-compensating cations, resulting in the emergence of acidic sites. The negatively charged skeletal oxygen near these cations acts as a basic site, interacting with the permanent quadrupole moment in $CO_2$ [35], leading to higher $CO_2$ adsorption rates. Consequently, in conjunction with the sorption data, it was determined that continuous sorption testing at 0 °C was most suitable for various temperature conditions. In prior studies, gangue has been employed as a raw material for $CO_2$ adsorbent synthesis, offering cost reductions and substantial environmental benefits. This suggests the potential utility of gangue-based 13-X molecular sieves for $CO_2$ capture.

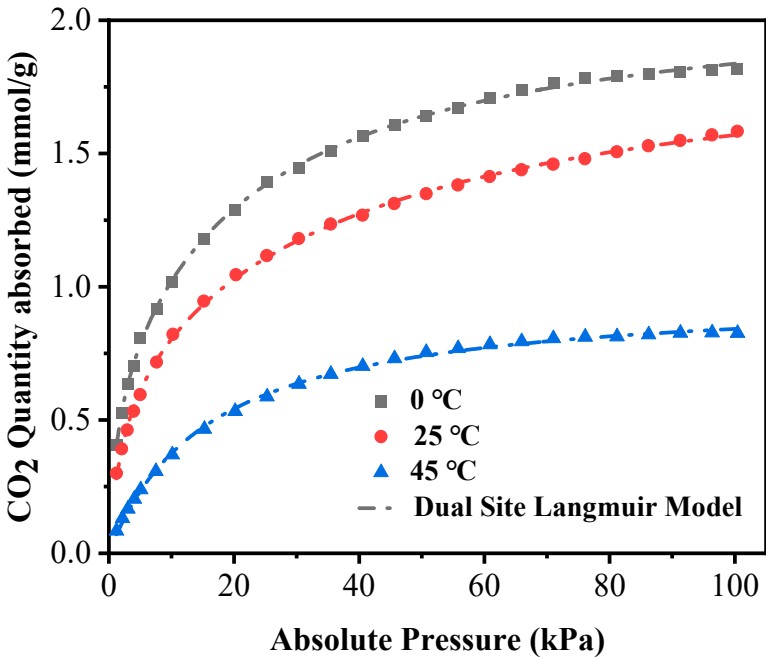

**Figure 7.** $CO_2$ adsorption isothermal fitting curves of synthesized 13-X molecular sieve Y9 at different temperatures and Dual-site Langmuir Model fitting.

The effect of synthesized 13-X molecular sieves on $CO_2$ adsorption performance at 25 °C was investigated at different adsorption pressures of 20 kPa, 40 kPa, 60 kPa, 80 kPa, and 100 kPa, respectively, as shown in Figure 8, which is the histogram of $CO_2$ adsorption capacity at different adsorption pressures. It can be observed from the figure that the adsorption capacity of 13-X molecular sieves for $CO_2$ showed an increasing trend with the increase of pressure. The adsorption capacity of the 13-X molecular sieve for $CO_2$ increased from 1.04 mmol/g to 1.27 mmol/g for pressure from 20 kPa to 40 kPa and

increased continuously with the pressure, and reached 1.58 mmol/g for the adsorption capacity at 100 kPa. The $CO_2$ adsorption capacity of the 13-X zeolite showed a tendency to increase and then decrease with the increase of the pressure. This indicates that the larger the adsorption pressure, the greater the influence the 13-X molecular sieve has on the adsorption capacity, and this also shows a positive correlation trend. Secondly, when the pressure increases, the asymmetric vibration of Si-O and Al-O tetrahedra in the molecular sieve will be strengthened, which makes the asymmetric vibration of Si-O and Al-O tetrahedra in the molecular sieve more obvious. The gas and water will be discharged from the pores, thus enlarging the diameter of the pores and improving the adsorption capacity of the 13-X molecular sieve for $CO_2$ gas.

**Table 4.** Isothermal model coefficients of $CO_2$ adsorption at different temperatures.

| Temperature (°C) | Langmuir Model | | | Dual-Site Langmuir Model | | | | |
|---|---|---|---|---|---|---|---|---|
| | $q_A$ (mmol/g) | $b_A$ (kPa$^{-1}$) | $R^2$ | $q_A$ (mmol/g) | $b_A$ (kPa$^{-1}$) | $q_B$ (mmol/g) | $b_B$ (kPa$^{-1}$) | $R^2$ |
| 0 | 1.91 | 0.13 | 0.9799 | 1.48 | 0.04 | 0.64 | 0.84 | 0.9995 |
| 25 | 1.65 | 0.10 | 0.9796 | 1.29 | 0.03 | 0.65 | 0.48 | 0.9994 |
| 45 | 0.98 | 0.06 | 0.9988 | 0.49 | 0.06 | 0.49 | 0.06 | 0.9988 |

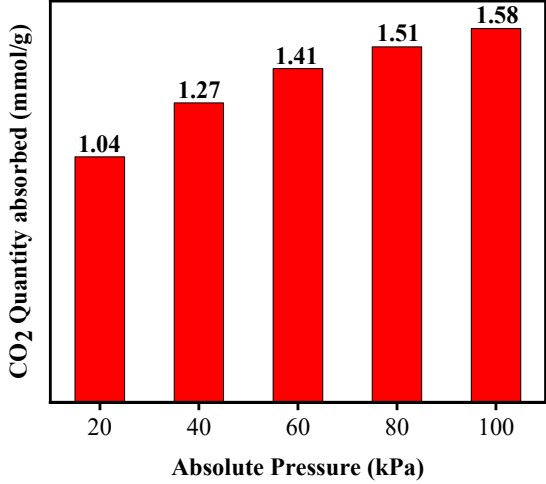

**Figure 8.** Column chart of $CO_2$ adsorption capacity under different adsorption pressures.

In order to examine the $CO_2$ adsorption capacity of competing adsorbents, the $CO_2$ capture capacity of the synthesized 13-X molecular sieve was compared with other $CO_2$ adsorbents reported in the literature in this study, and the results of the comparison are shown in Table 5. As can be seen from the data in Table 5, the $CO_2$ capture capacity of the 13-X molecular sieve synthesized in this work is close to that of adsorbents reported in the literature [39] and exceeds that of many reported $CO_2$ adsorbents, such as Hierarchical melamine resin sponges [40] and Amine-impregnated Silica foam [41].

**Table 5.** Comparison of $CO_2$ trapping capacity of different adsorbents.

| Sample | Adsorption Temperature (°C) | Adsorption Capacity (mmol/g) | References |
|---|---|---|---|
| zeolite 13-X | 0 | 1.8 | Present work |
| Hierarchical melamine resin sponges | 0 | 1.7 | [40] |
| β-zeolite | 30 | 1.8 | [39] |
| Monolithic Ni/ZSM-5 | 35 | 2.4 | [42] |
| Amine-impregnated Silica foam | 25 | 1.48 | [41] |
| ZSM-5 | 0 | 2.37 | [43] |

## 4. Conclusions

The 13-X molecular sieve porous materials were synthesized by alkali melting and hydrothermal treatment using cheap and readily available solid waste-based coal gangue as a raw material, without adding any silicon or alumina sources. The effects of Si/Al, alkalinity, crystallization temperature, and hydrothermal time on the synthesis of the 13-X molecular sieve were investigated using the orthogonal experimental design. The prepared 13-X molecular sieve exhibited a highly crystalline pore structure with a BET-specific surface area of 377.02 $m^2$/g, and it showed excellent $CO_2$ capture capacity ($CO_2$ adsorption capacity of 1.82 mmol/g) at 0 °C and 1 bar. The low cost and high value-added availability of coal gangue and the high $CO_2$ capture performance of coal gangue-based 13-X molecular sieves indicated that the synthesis of 13-X zeolite for $CO_2$ capture from gangue is an ideal resource utilization method for coal gangue that can be used in the future.

**Author Contributions:** Investigation, Y.L.; Resources, Y.G.; Data curation, S.L., B.X., H.H. and L.K.; Writing—original draft, D.Y.; Writing—review & editing, H.D. All authors have read and agreed to the published version of the manuscript.

**Funding:** This research was supported funded by the State Key Laboratory for Mechanical Behavior of Materials (Grant No. 20232508). the Excellent Youth Science and Technology Fund Project of Xi'an University of Science and Technology (Grant No. 6310221009), the National Natural Science Foundation of China (No. 52172099), the Basic Research Plan of Natural Science of Shaanxi Province (No. 2020JQ-754).

**Institutional Review Board Statement:** Not applicable.

**Informed Consent Statement:** Not applicable.

**Data Availability Statement:** Data will be made available on request.

**Conflicts of Interest:** The authors declare no conflict of interest.

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
