# Peer review of "Study on Green Controllable Preparation of Coal Gangue-Based 13-X Molecular Sieves and Its CO2 Capture Application"

_coatings, doi:10.3390/coatings13111886_

Round 1
Reviewer 1 Report
Comments and Suggestions for Authors
In this study, gangue-based 13-X molecular sieve porous materials were synthesized and studied for the adsorption of CO2. I appreciate the well-written paper, but the novelty of the work is unclear and a few issues need to be addressed in accordance with the following comments:
- The novelty of the work is unclear.
- The purity of chemicals should be included.
- Why was an orthogonal experimental design chosen for this study?
- How were the influencing factors and their levels chosen? For instance, it should be clear why the hydrothermal times have been chosen to be 8, 10, and 12. And so on.
- Despite the higher R2 than in the Dual-site Langmuir model, it appears that the maximum adsorption capacity obtained from the Langmuir model is closer to the experimental value than the Dual-site model. So, I think the isotherms follow the Langmuir model.
- Adsorption mechanisms should be discussed.
- There should be a clear discussion of the effects of the factors listed in Table 2.
Author Response
Dear reviewer, since it was the first time we contributed to your journal, the process was not very clear, resulting in multiple submissions, we hereby express our sincere apologies. For your review comments, please refer to the attachment submitted this time. We would like to express our sincere apologies again, and wish you a happy life.

Reviewer 2 Report
Comments and Suggestions for Authors
In this article, the authors tried to synthesize a novel low-cost 13-X CO2 adsorbent which is promising and deserved a great appreciation. A lot of work has been done and a comprehensive study has been prepared; but still there are some concerns about this study which shall be cleared.
1.The English language quality must be rechecked and revised. There are many grammatical errors through the study.
2.I think it is better to categorize the gangue-based adsorbent within the zeolites, not in molecular sieve type.
3.Please indicate the pros and cons of all mentioned adsorbents in a table and then, mention the reason of selecting molecular sieve (zeolite) between them.
4. In page 2, paragraph 2, line 63: you mentioned thesis instead of “study”!
5. The section “2.2.2. CO2 adsorption measurement” is related to section “2.2.1. Synthesis of gangue-based 13-X molecular sieve”; so, this paragraph should be added to the end of the section 2.2.1.
6.In the section 2.2.2., you should discuss about the set up you used, experiment duration, and some other parameters important in adsorption process.
7.The section 3.2. title is better to change into: “Evaluation of the synthesized 13-X zeolite”
8. The section 3.1.2. is really vague! What is really your dependent variable?? How did you apply the “Orthogonal experimental design” to evaluate your dependent variable??
Actually, you defined and indicated a figure (Fig. 3) instead of a certain variable and then concluded all results from that only one figure!! I think there is a serious mistake here.
9.In section 3.2.2., why did you select Y7??
10.As you mentioned, your observed CO2 capacity was 1.80 mmol/g; while in many studies, researchers have reported much higher quantities. For instance, you can compare the capacities reported in the following studies with yours:
· Xia, Y., Mokaya, R., Walker, G. S., & Zhu, Y. (2011). Superior CO2 adsorption capacity on N‐doped, high‐surface‐area, microporous carbons templated from zeolite. Advanced Energy Materials, 1(4), 678-683.
· Jadhav, P. D., Chatti, R. V., Biniwale, R. B., Labhsetwar, N. K., Devotta, S., & Rayalu, S. S. (2007). Monoethanol amine modified zeolite 13X for CO2 adsorption at different temperatures. Energy & Fuels, 21(6), 3555-3559.
· Pham, T. H., Lee, B. K., & Kim, J. (2016). Novel improvement of CO2 adsorption capacity and selectivity by ethylenediamine-modified nano zeolite. Journal of the Taiwan Institute of Chemical Engineers, 66, 239-248.
· Karimi, M., Shirzad, M., Silva, J. A., & Rodrigues, A. E. (2023). Carbon dioxide separation and capture by adsorption: a review. Environmental Chemistry Letters, 1-44.
In this way, how can you justify the lower CO2 capacity in your work?
Comments on the Quality of English LanguageThe English language quality must be rechecked and revised. There are many grammatical errors through the study.
Reviewer 3 Report
Comments and Suggestions for Authors
The authors investigated the effectiveness of 13-X molecular sieve porous materials synthesized using solid waste coal gangue for CO2 capture.
Some comments for improvement are as follows:
General:
The title should be revised to better reflect the work. What is the proof that the synthesis is low-cost and efficient for CO2 capture?
Introduction:
“According to relevant data records, the concentration of CO2 in the atmosphere is increasing even when moving in indoor spaces…”. What is the significance or relevance of relating to the indoor space?
“…..adsorption process technology is of great interest due to its convenience versatility and applicability.” What sort of convenience versatility and applicability that is referred here? How about when the adsorber is saturated?
Methodology:
For the materials, it would be better to state the purity of the chemicals used in percentage value.
The CO2 adsorption measurement section is unclear. What was the set-up used for CO2 adsorption measurement?
“The specific surface area and pore size distribution of the samples were obtained using the N2 adsorption-desorption isotherm at 77 k of liquid nitrogen temperature using Micromeritics ASAP 2020 analyzer”. What is the 77 k referring to?
“The CO2 gas adsorption-desorption isotherms of Y9 were measured on a Quantachrome version 5.21 gas adsorption analyzer.” What is Y9 referring to?
Results and discussion:
Some part of the results and discussion are actually methodology. Separate them accordingly.
What is the proof that “the addition of alkali source can effectively break the inert crystal structure of quartz in gangue”?
“Different NaOH led to other types of zeolites, respectively, which showed to be very effective for
the extraction of Si and Al from gangue and the formation of new zeolites”. What kind of different NaOH was used? Please clarify.
Quite a number of discussions is without proper supporting proof.
The so-called efficient adsorption capacity is measured based on 0 degree C of adsorption temperature. What application would be of such temperature?
Comments on the Quality of English LanguageThis manuscript needs to be proofread by a native English speaker for better clarity.
Reviewer 4 Report
Comments and Suggestions for Authors
The authors studied the synthesis of 13-X molecular sieve porous materials using solid waste coal gangue as a source of silicon and aluminum towards CO2 capture. In particular, they have shown that 13-X molecular sieve material exhibit a porosity with uniform pores and complete crystalline morphology, being its CO2 adsorption capacity at 0 °C of 1.8165 mmol/g. Although the authors conducted studies on the evaluation of the synthetized 13-X molecular sieve material, some characterization and discussion are missed, being completely necessary to publish in this journal. Below are some comments from me as a reviewer. I would be very happy to receive responses to them.
1. English and writing skills are of utmost importance to properly write a scientific article. It was really hard to read and understand the content of the manuscript. I highly recommend carefully and thoroughly reviewing the grammatical content of the whole manuscript.
2. An explanation about the objective of doing a pretreatment to the raw coal gangue powder with NaOH at 750 ºC is missed. As well as the choice of such specific conditions (mass ratio of 1:0,6 and 750 ºC for 4 h) and atmosphere conditions of the heat-treatment (it is mentioned that the sample is “calcined” in SEM-EDS analysis in Page 6). Moreover, I think that “roasted” is not the proper word to describe the process.
3. Figure 2 should appears colorful, just like the other figures.
4. Particle size distribution is mentioned in Page 4, while no data related to that is shown.
5. There is only one FESEM micrograph, while it is written “micrographs”. Moreover, it should be indicated in which figure it appears.
6. XRD pattern of the gangue (Figure 2) should be also included in Figure 3 for comparison purpose, since it is discussed in the text.
7. It is affirmed that “the glass body within the raw material system has appeared dissolved” in Page 5. Which glass body are the authors taking about? It is not mentioned before and the raw coal gangue is highly crystalline (Figure 2). I recommend to modify that sentence.
8. Moreover, in the same paragraph, the presence of different peaks in samples Y1-6 are associated to silicon-aluminate phases. Therefore, corresponding XRD patterns should be included.
9. Authors indicate that the Si/Al molar ratio is a key factor for the formation of different types of molecular sieves, observing the standard 13-X molecular sieve values when the highest ratio (Si/Al molar ratio of 6) is employed. What about if the Si/Al molar ratio is continues to increase? The same question comes to me after read the conclusions related with the effect of NaOH solutions. What about if the concentration of NaOH continues to decrease? What about a lower temperature (<80 ºC)? The considered as the best conditions for the synthesized 13-X molecular sieve cannot be considered the optimal, since additional experimental conditions are needed.
10. The reason to study the sample Y7 by SEM-EDS analysis is missed.
11. EDS spectrum is not clear, as well as the elements quantification from EDS of the raw material gangue and Y7 should be also provided in the text. Moreover, the sum of the four elements shown (Na, Al, Si and O) is 73.27 %. What about the rest?
12. The BET specific surface area, as well as the pore size distribution and the CO2 trapping capacity of the commercial 13-X molecular sieve should be included and discussed for comparison purpose.

English and writing skills are of utmost importance to properly write a scientific article. It was really hard to read and understand the content of the manuscript. I highly recommend carefully and thoroughly reviewing the grammatical content of the whole manuscript.
Round 2
Reviewer 1 Report
Comments and Suggestions for Authors
The paper can be accepted in this version
Author Response
Dear reviewer, thank you for taking time out of your busy schedule to review our responses. I am very glad to cooperate with you to make this article better. I wish you good health and a happy lifeReviewer 2 Report
Comments and Suggestions for Authors
Comment 6: In table 2, you must mention all references used in each row.
If the data used in this study are not original, how you can claim that the adsorbent has been firstly introduced by you??
Comment 8 is still not justified correctly
Comment 9: all the justification shall be mentioned in the manuscript with relevant references.
Comments on the Quality of English LanguageEnglish language of this study needs a moderate revision.
Reviewer 3 Report
Comments and Suggestions for Authors
The concerns have been addressed.
Author Response
Dear reviewer, thank you for taking time out of your busy schedule to review our responses. I am very glad to cooperate with you to make this article better. I wish you good health and a happy life
Reviewer 4 Report
Comments and Suggestions for Authors
Thank you for responding to all my comments and suggestions.
Now, the manuscript is of sufficient quality to be published in this journal.
Author Response

(The authors gave the same response as above.)
